# Beyond Endocasts: Using Predicted Brain-Structure Volumes of Extinct Birds to Assess Neuroanatomical and Behavioral Inferences

**Catherine M. Early** [1,*,†] **, Ryan C. Ridgely** [2] **and Lawrence M. Witmer** [2]

[1] Department of Biological Sciences, Ohio University, Athens, OH 45701, USA
[2] Department of Biomedical Sciences, Heritage College of Osteopathic Medicine, Ohio University, Athens, OH 45701, USA; ridgely@ohio.edu (R.C.R.); witmerL@ohio.edu (L.M.W.)
[*] Correspondence: cmearly1311@gmail.com
[†] Current Address: Florida Museum of Natural History, University of Florida, Gainesville, FL 32611, USA.

**Abstract:** The shape of the brain influences skull morphology in birds, and both traits are driven by phylogenetic and functional constraints. Studies on avian cranial and neuroanatomical evolution are strengthened by data on extinct birds, but complete, 3D-preserved vertebrate brains are not known from the fossil record, so brain endocasts often serve as proxies. Recent work on extant birds shows that the Wulst and optic lobe faithfully represent the size of their underlying brain structures, both of which are involved in avian visual pathways. The endocasts of seven extinct birds were generated from microCT scans of their skulls to add to an existing sample of endocasts of extant birds, and the surface areas of their Wulsts and optic lobes were measured. A phylogenetic prediction method based on Bayesian inference was used to calculate the volumes of the brain structures of these extinct birds based on the surface areas of their overlying endocast structures. This analysis resulted in hyperpallium volumes of five of these extinct birds and optic tectum volumes of all seven extinct birds. Phylogenetic ANCOVA (phyANCOVA) were performed on regressions of the brain-structure volumes and endocast structure surface areas on various brain size metrics to determine if the relative sizes of these structures in any extinct birds were significantly different from those of the extant birds in the sample. Phylogenetic ANCOVA indicated that no extinct birds studied had relative hyperpallial volumes that were significantly different from the extant sample, nor were any of their optic tecta relatively hypertrophied. The optic tectum of *Dinornis robustus* was significantly smaller relative to brain size than any of the extant birds in our sample. This study provides an analytical framework for testing the hypotheses of potential functional behavioral capabilities of other extinct birds based on their endocasts.

**Keywords:** endocast; neuroanatomy; brain; bird; evolution

---

## 1. Introduction

Our deep fascination with extinct animals such as dinosaurs is driven not by their dry bones that remain with us today but by thinking of them as living and breathing animals, interacting with each other and their environments. These interactions are determined by the various behaviors in which organisms engage, which are in turn enabled by the functional capabilities of their brains. The brain-behavior connection in extant and extinct organisms has led to rapid growth of the field of comparative neuroanatomy [1–3]. In this field, correlations are drawn between the size of a brain structure in a species of extant animal and the relative emphasis on the behavior it mediates. Efforts to use these correlations to infer functional capabilities in extinct animals have been stymied by the fact that whole soft-tissue organs, including the brain, do not preserve in three dimensions in the fossil

record. The fossil record is much better at preserving hard tissues such as bone, and fortunately, the brain is surrounded by bone to varying extents. The space within these bones constitutes the endocranial cavity, which can then serve as a mold to generate a brain endocast (or simply "endocast" for the remainder of this article). These can be physical endocasts (e.g., latex; [4]) or, more commonly in recent years, virtual endocasts reconstructed from computed tomography (CT), a non-destructive sampling technique [5]. The endocast can be a very accurate representation of the size and shape of the external surface of the brain in birds [6].

The fidelity of the avian endocasts to their brains, and the fact that the endocast is the most direct evidence of neuroanatomy available from the fossil record, has led many researchers to make inferences about behavior from the endocasts of the few extinct birds with three-dimensionally preserved skulls. Given the correlations found with higher-order behavioral traits associated with intelligence and relative brain volume (summarized in [7]), encephalization (i.e., the size of the brain relative to the body) has been a major topic of interest in endocast studies [8–10]. The overall shape of the endocast and how that shape correlates with behavioral and life-history traits have become more available for analysis with the advent of 3D geometric morphometric techniques [11–14]. The most common approach in endocast studies has been to describe a fossil endocast in comparison with its living relatives, both in terms of general morphology and relative size of discrete structures [7,9,11,15–26]. These studies have allowed researchers to document trends in avian brain evolution by using endocast morphology as a proxy for neuroanatomy. Often, they link the relative size of an endocast structure to the importance of the function mediated by its underlying brain structure in the extinct species as compared to its extant relatives.

One of the endocast structures frequently used to infer behavior or functional capabilities is the sagittal eminence, or Wulst (Figure 1). Its underlying brain structure, the hyperpallium, is the target of the thalamofugal visual pathway, the other visual pathway in birds. The thalamofugal visual pathway has been linked to perception of illusory contours [27], pattern discrimination at a distance [28], higher-level motion integration [29], spatial orientation [30], and visual magnetoreception [31,32]. Although portions of the hyperpallium are also involved in somatosensation and motor control, it is still thought that the majority of this structure is devoted to integrating visual information [33]. As a result, the size of the Wulst on an endocast is often interpreted as correlating with visual capabilities in extinct birds. Similar to the optic lobe, the expansion of the Wulst (and presumably, its underlying brain structure) is a characteristic trait of avian brain evolution [7,10,34]. Many studies on the endocasts of extinct taxa have found Wulsts that fall within the range of variation seen in extant birds and thus have ascribed them similar visual capabilities [11,18–26]. The trend in Wulst size observed in extinct and extant penguins fits the general avian pattern of Wulst expansion throughout evolution of the clade [21,23,25]. Counter to this trend, the Wulst seems to have been larger in extinct columbiforms than their extant relatives [24].

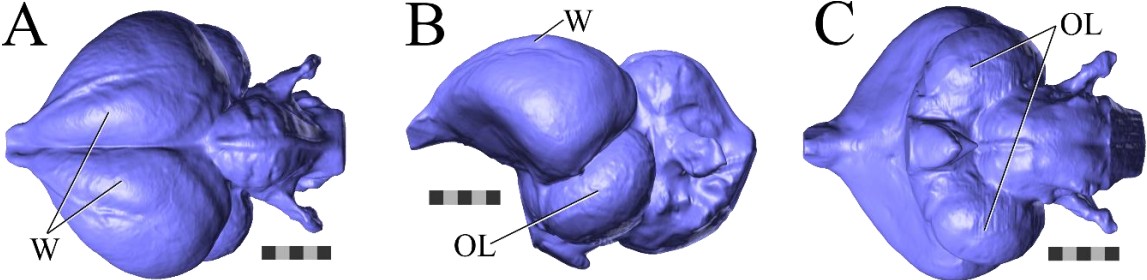

**Figure 1.** The endocast of an extant bird, *Bonasa umbellus* (AMNH Birds SKEL 21616), in (**A**) dorsal, (**B**) lateral, and (**C**) ventral views showing the location of the Wulst and optic lobe on an avian endocast. For all views, rostral is to the left. Abbreviations: OL, optic lobe; W, Wulst. Scale bars = 5 mm.

Another endocast structure that has been a major focus of attention is the optic lobe, which overlies the bilateral expansion of the midbrain that serves as a relay point for sensory information heading toward the forebrain for higher-level processing (Figure 1). The majority of this midbrain expansion is occupied by the optic tectum and optic tract, which receive visual information from the retina as part of the tectofugal visual pathway [35]. The tectofugal visual pathway is considered the major visual pathway in birds as it receives over 90% of the retinal projections [36–38], and it is involved in discrimination of brightness, color, and pattern, processing simple and complex motion, and selective attention [39–45]. Because of the superficial position of the optic tectum relative to the other midbrain nuclei, the optic lobe of the endocast is equivalent to the externally visible portion of the optic tectum and optic tract, so the lobe's size is often interpreted as an indicator of visual abilities. Its expansion and ventrolateral displacement by the expanded telencephalic hemispheres is one of the hallmark traits of brain evolution on the lineage from non-avian theropods to birds [7,46,47]. This trend is supported by the many fossil endocasts whose optic lobes appear similar in size to those of their extant relatives [11,16–21,23,24,48]. There are cases in which the optic lobes of some extinct birds appear to be relatively smaller than those of their extant relatives, which has been interpreted as a possible indication of reduction in visual abilities [9,22,26,49–51]. In keeping with the finding that the optic tectum is not particularly hypertrophied in any of the extant avian clades sampled by [52], there are no documented cases in the fossil record of birds with relatively larger optic lobes.

Inherent in these inferences of functional capabilities based on Wulst or optic lobe size has been an untested assumption: the size of an endocast structure is representative of the size of the underlying brain structure, and thus any behavioral inferences that can be made from brain-structure size can also be made from endocast structure size. This shortcoming was recognized by some authors who attempted to address it by using homologous osteological landmarks to partition endocasts into equivalent regions for comparison, but these same authors recognized that this method could not capture internal organization [10,53,54]. Another method that was advocated to establish relationships between external structures preserved on the endocast and internal brain structures was to determine the extent to which exposed surface area can predict the deeper volume [55]. This aim was met by our previous study [51] in which the volumes of the hyperpallium and optic tectum, two brain structures involved in the visual pathways of birds, were regressed on the surface areas of the Wulst and optic lobe, respectively, which are the endocast structures overlying those brain structures. These regressions demonstrated a strong, significant correlation between the volumes of the brain structures and the surface areas of their overlying endocast structures, indicating that the sizes of the endocast structures can be used as a proxy for the sizes of their associated brain structures.

The correlation between the surface area of the Wulst and the volume of the hyperpallium supports the tendency of other authors to equate the two structures when making functional inferences [18–21,23–25,48]. Similarly, the correlation between the surface area of the optic lobe and the volume of the optic tectum lends credence to the inferences of potential functional capabilities made in previous publications that were based on the size of the optic lobe [18,22,26,50]. However, none of these publications used quantitative approaches to determine if the relative size of endocast structures apparent from visual appraisal of the endocasts translates to true expansion or reduction of the brain region. Phylogenetic ANCOVA (phyANCOVA) allow for a direct test of significance of a single species' deviation from the allometric trend of a group [56], but this approach has not been taken with avian endocasts. Such a test is beneficial when dealing with fossils as there is often only one species available for a given genus or higher taxon. These newly developed methods will allow incorporation of extinct birds into analyses of hyper- or hypotrophy of brain structures like the hyperpallium and optic tectum.

In this study, we leverage the relationships established between Wulst surface area and hyperpallium volume as well as optic lobe surface area and optic tectum volume from the sample of extant birds in an earlier study [51] to predict the volumes of the brain structures from the surface areas of the associated endocast structures of a few extinct birds. These predicted brain-structure volumes

(hyperpallium or optic tectum volume) were combined with actual brain-structure volumes of extant birds. These combined datasets of hyperpallium or optic tectum volumes were regressed on various measures of brain size such as foramen magnum area or brain volume minus brain-structure volume (traditionally termed "brain-rest volume," although here it will be referred to as "brain-remainder volume"). We performed phyANCOVA on these regressions to determine if any of the extinct birds had brain-structure volumes that were relatively larger or smaller than those of their extant relatives. The same analyses were performed on regressions of endocast structure surface areas of extinct and extant birds on foramen magnum area and endocast surface area minus endocast structure surface area to further explore the utility of endocasts alone as representatives of neuroanatomy. The fossils analyzed represent case studies rather than a comprehensive sample and serve to demonstrate the applicability of this method to endocasts of other extinct birds.

## 2. Materials and Methods

A broad comparative sample of the endocast-structure surface areas and brain-structure volumes of extant birds was compiled for a previous study [51]. Analyses of this sample demonstrated a strong, significant correlation between the size of the endocast structures and their underlying brain structures, indicating that it would be appropriate to use this sample to calculate brain-structure volumes in extinct taxa. Five of the endocasts of extinct birds in the present study were complete enough to allow prediction of hyperpallium volumes, and optic tectum volumes were predicted for all seven extinct birds (Table 1, Figure 2). Many of these specimens had been scanned as part of other studies [21,47,49,50,57–59] and their data and endocasts were contributed to this case study. This sample of endocasts of extinct birds was determined by the availability of the specimens for study and a high degree of 3D preservation with minimal deformation. As such, the present sample is determined largely by the nature of the fossil record.

**Table 1.** The fossil specimens used in this study, institutions at which they were scanned, reconstruction steps taken, and publications in which the scan data or endocasts were presented previously.

| Species | Specimen | CT Facility | Reconstruction Steps | Publication |
|---------|----------|-------------|---------------------|-------------|
| *Archaeopteryx lithographica* | NHMUK PAL PV OR 37001 | UTCT | Mirrored left hemisphere | [17] |
| *Lithornis plebius* | USNM PAL 336534 | OUμCT | Interpolated between two pieces | [57] |
| *Dinornis robustus* | FMNH PA 35 | OhioHealth O'Bleness Hospital | None | [49,50] |
| unnamed Miocene galliform | AMNH FR 8629 | AMNH Microscopy and Imaging Facility | Patched holes | Ksepka et al. in prep. |
| *Paraptenodytes antarcticus* | AMNH FR 3338 | Stony Brook University Hospital | Patched holes | [21] |
| *Psilopterus lemoinei* | AMNH FR 9257 | OUμCT | Patched holes | [58,59] |
| *Llallawavis scagliai* | MMP 5050 | Instituto Radiológico | Patched holes | [58,59] |

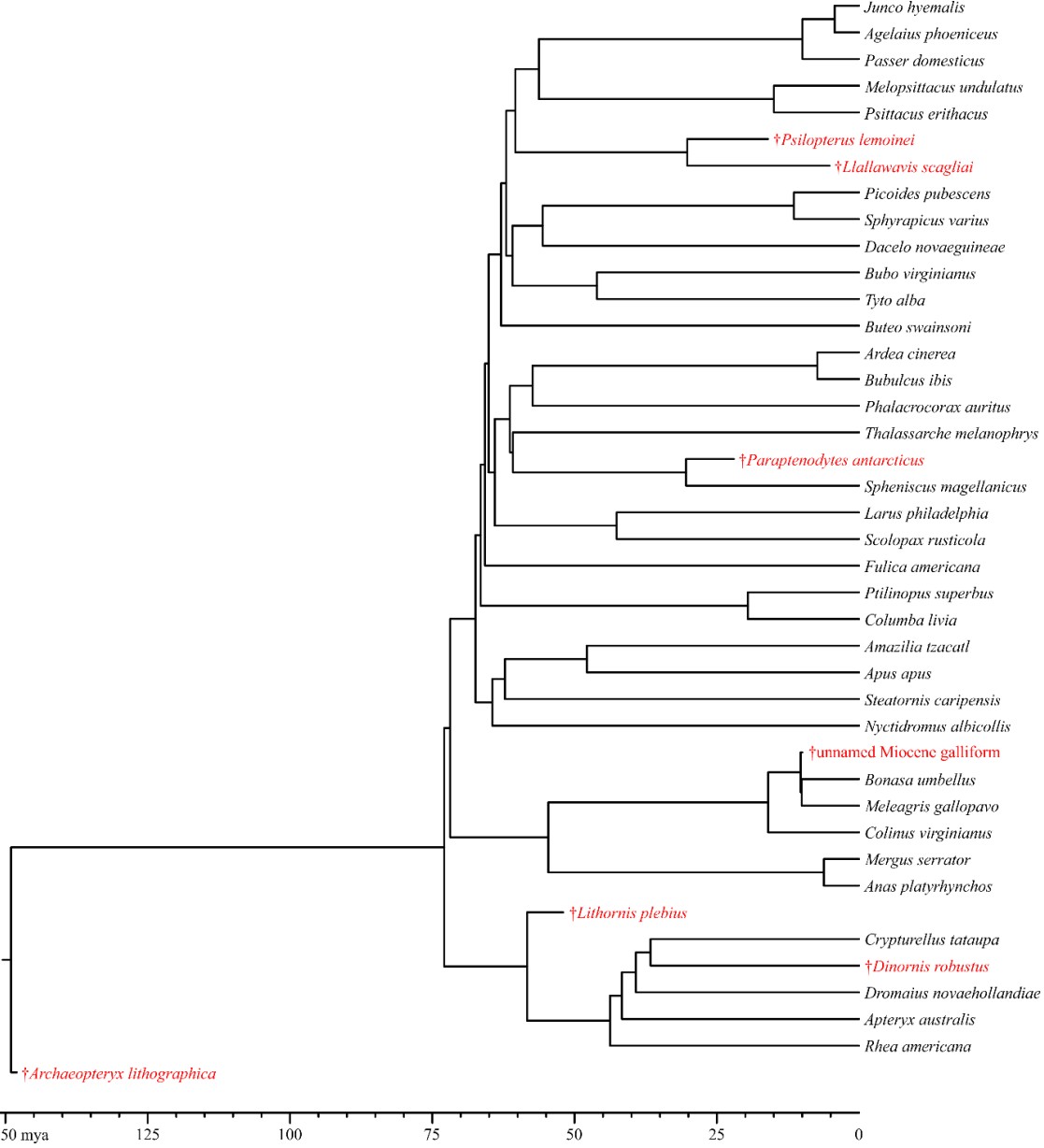

**Figure 2.** A phylogeny of the subset of species for which endocasts of a representative specimen were analyzed in the present study. The seven extinct birds are in red, and the timescale is in millions of years. For ease of visualization, this tree was subsampled from a tree that includes all species listed in Table S1. That tree is a composite of the backbone of Prum et al. [60] and the finer-level topology of Jetz et al. [61]. The complete tre file, which includes divergence time estimates, is available in the Supplementary Materials (S4).

Endocasts of seven extinct birds were included in this study (Table 1). The endocasts of the terror birds *Llallawavis scagliai* (MMP 5050) were segmented previously by F. J. Degrange, and his Avizo label fields were generously contributed to the present study. Based on qualitative comparisons, Degrange et al. [58,59] found that *Llallawavis* and *Psilopterus* had well-developed Wulsts and optic lobes like their extant cariamiform relatives. The endocast of the early Miocene stem penguin *Paraptenodytes antarcticus* (AMNH FR 3338) was published by Ksepka et al. [21], and the Avizo label fields used to segment the endocast were generously contributed. As with the terror birds, Ksepka et al. [21] found that the relative size of its Wulst and optic lobe were in line with those of extant penguins. Scan data of the braincase of *Archaeopteryx lithographica* (NHMUK PAL PV OR 37001)

were generously provided by A. Milner to L. M. Witmer in 2004, and, although multiple versions of the endocast of this specimen have been published [10,17,47], the endocast analyzed in the present study was segmented by us. The endocast of *Lithornis plebius* (USNM PAL 336534) was also segmented by us and published as part of an earlier study [57]. The endocast of *Dinornis robustus* (FMNH PA 35) was segmented by C. M. Early for this and other studies [49–51]. Scan data of the unnamed Miocene galliform (AMNH FR 8629; [62]) were contributed by a collaborator and segmented by Early for a forthcoming description and analysis of this specimen, its phylogenetic position, and its neuroanatomy.

The skulls of each specimen were scanned at slice thicknesses ranging from 41 μm to 350 μm, as determined by the size of the specimen, on various CT scanners (see Table 1). Different versions of the 3D visualization program Avizo (Thermo Fisher Scientific, Waltham, MA, USA) were used to process all scan data. Avizo used the metric value of each voxel as recorded by the scanner, allowing quantitative measurements to be generated. Following best practices outlined by a working group of avian brain endocast researchers [54], the voxels of the CT scans corresponding to the endocranial spaces were isolated in Avizo and assigned to a material using the program's "magic wand," "paint," and "select" tools. The virtual model produced by this workflow represents the endocast of the specimen of interest.

As was done with the segmentation of endocasts of extant birds in our previous study [51], attention was paid to the removal of endocranial vasculature and cranial nerves from the endocast along the boundaries where these structures intersected each other. The points of flexure between the vasculature or nerves and the endocast in three different views were used as indicators of the boundary between these structures and the endocast. Endocranial vasculature with clear boundaries such as the carotid artery or the transversotrigeminal vein was removed using these boundaries, but the occipital sinus often manifests as a diffuse structure and thus was not removed in any taxa. Unlike vasculature, nerves do comprise part of the nervous system of birds, but they were treated as separate entities in this study, so their canals were also removed from the endocasts. These removals were performed as consistently as possible to ensure that the vasculature and nerves would not change the estimates of overall endocast surface area or volume. The fossa and canals of the trigeminal nerve (cranial nerve [CN] V) could directly impact endocast structure surface area measurements because these structures cover the ventral surface of the bony fossa tecti mesencephali in many birds. This relationship results in CN V being continuous with the optic lobe on the endocast, so its consistent removal was key to yielding accurate surface area values for the optic lobes. As in the earlier study on extant taxa [51], the ventrolateral curvature of the fossa tecti mesencephali and the angle of the brainstem in axial view guided removal of CN V. The boundary was then smoothed in sagittal slices of CT scan data using an interpolation-based approach.

Because of the often fragmented state of fossils, almost all of the endocasts of extinct birds used in this sample demonstrate levels of damage not seen on the endocasts of extant birds (Figure 3). In cases of minor breakage, best-practices approaches were used to smooth the breaks and connect the separated portions of the endocasts [54]. In the case of *Archaeopteryx*, one half of the endocast was well-preserved but the other half was crushed, so we mirrored the well-preserved half at the midline to yield a complete endocast (Figure 3A). The specimen of *Lithornis* was so fragmented that we used Avizo to make a composite endocast from CT scans of two fragments of the skull of the same specimen, interpolating to fill the space between the specimens (Figure 3B). Details of treatments for each specimen are provided in Table 1. Unfortunately, damage to the skull roof often affected the Wulst and made it unmeasurable, so fewer extinct taxa were available for that analysis. For example, *Lithornis* seems to have a Wulst (Figure 3B), but the deformation of its endocast has rendered the boundaries of the Wulst too poorly defined for measurement. The structure on the endocast of *Archaeopteryx* that has been proposed as a potential homolog for the Wulst [10] was also not measured for this reason and because of its unresolved identity (see Discussion). All of the endocasts available for this study had at least one well-preserved optic lobe, and in the cases where only one was preserved, its measured surface area was doubled. The fragmented nature of most of the fossils in our sample does mean that

overall endocast surface areas or volumes should be evaluated with caution, as overlap of bones could cause truncation of the endocast and a reduction in these values.

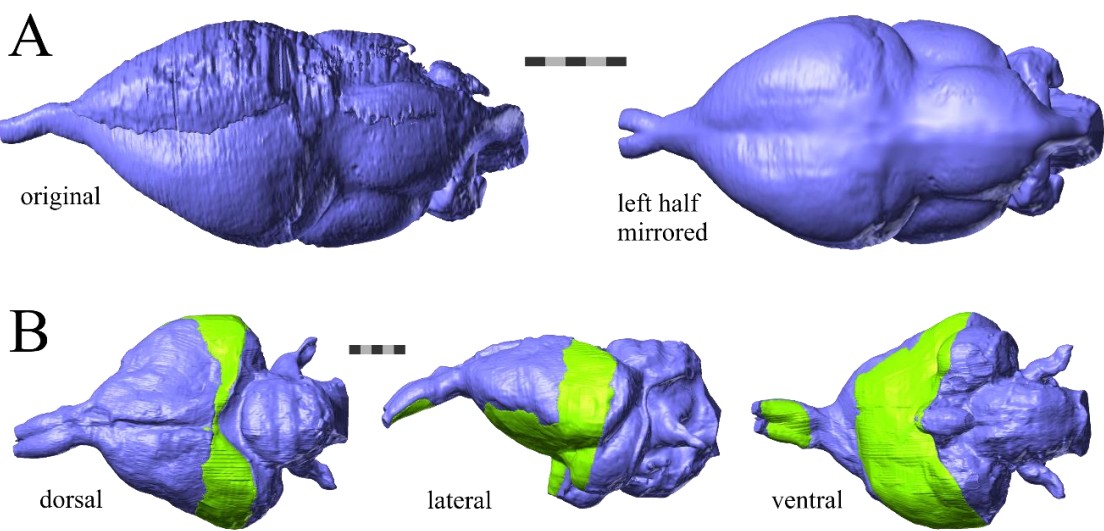

**Figure 3.** Examples of varying degrees of artifactual damage and deformation represented in a subsample of fossil endocasts. (**A**) The original endocast of *Archaeopteryx lithographica* (NHMUK PAL PV OR 37001) (left) compared to the endocast that was generated by mirroring the left half of the endocast on the midline (right); both endocasts are in dorsal view. (**B**) The endocast of *Lithornis plebius* (USNM PAL 336534) in three views with preserved portions of the endocast shown in blue and the interpolated portions created to span the gap in green. Scale bars = 5 mm.

The methods used to measure endocast structure surface areas in this study were similar to previous studies [20,26,51]. Each endocast was reduced to a more manageable size, converted to an STL file, and then imported into the 3D modeling software Maya (Autodesk, San Rafael, CA, USA). All faces of the mesh were assigned to a baseline "endocast remainder" material. Maya's "paintbrush" tool was used to select the Wulst using the bony boundaries of the crista frontalis interna medially, the crista vallecularis laterally, the point at which the crista vallecularis and crista frontalis interna met rostrally, and the subtle inflection between the Wulst and the impression for the hippocampus caudally. These faces were subtracted from the "endocast remainder" material and added to a "Wulst" material. The same tool was used to select the optic lobes on the endocasts using the bony boundaries of the crista tentorialis dorsally, the inflection point between the fossa cranii caudalis and the ventral floor of the fossa tecti mesencephali (where CN V had been removed) ventrally, the impression of the tractus opticus rostrally, and the crista tecti caudalis caudally. The selected faces were subtracted from the "endocast remainder" material and added to an "optic lobe" material. The "Wulst," "optic lobe," and "endocast remainder" materials were exported from Maya, converted to STL files with MeshLab (Visual Computing Laboratory, Tuscany, Italy), and imported into Avizo where their surface areas were measured with the "surface area volume" feature. The resulting values are listed in Table S1.

The endocast-structure surface areas of the extinct taxa were added to the existing dataset of endocast-structure surface areas and brain-structure volumes of extant taxa from [51] (Table S1). Additional brain-structure volumes and overall brain volumes of extant taxa for which endocasts had not been generated in the previous study were gleaned from published studies and added to this dataset. Other authors have already established that the endocasts of birds closely match their brains in terms of volume [6], so the volumes of the endocasts of extinct birds in the sample were regarded as a proxy for brain volume. Brain- and endocast-structure size relative to brain and endocast size rather than to body size were analyzed to generate results that were as directly comparable as possible to existing histological studies on relative brain-structure size [52,63–71].

Given the uncertainties associated with endocranial volumes estimated from fossils, which vary in quality of preservation, it seemed prudent to explore other metrics against which scaling of brain-structure volumes could be assessed, such as the foramen magnum. Using foramen magnum area as a scaling factor for relative brain-structure size has been proposed for primates, another clade of highly encephalized vertebrates, but has not been broadly implemented in birds [72]. For this approach, the greatest width and height of the foramen magnum was measured with calipers or with Avizo's linear "Measurement" tool. These measurements on the two fossils that exhibited the most extensive deformation, *Archaeopteryx* and *Lithornis*, were supplemented with measurements from other endocasts. The previously published endocast of the London specimen of *Archaeopteryx* also employed the mirroring technique [17], so foramen magnum width and height were measured from the caudal view and these measurements were averaged with those taken from the newly generated endocast. The foramen magnum heights of *Lithornis celetius* (USNM PAL 290601) and *Lithornis promiscuus* (USNM PAL 391983) were recorded and averaged with that measured from the specimen used in this study as that specimen demonstrated significant breakage in the dorsal margin of the foramen magnum; the value used thus represents a mean for the genus, which is not ideal but is adequate pending discovery of more complete material. Foramen magnum area was calculated using the formula for the area of an ellipse. Foramen magnum width was also included as a potential scaling factor as it is easy to directly measure, and it was less impacted by deformation than foramen magnum height in the fossils in our sample. All measurements for extant and extinct taxa are available in Table S1.

All statistical analyses were run on log-transformed values and performed with R (R Foundation for Statistical Computing, Vienna, Austria [73]) running in RStudio (RStudio, Inc., Boston, MA, USA [74]). In our previous study on extant birds [51], the {ape} package [75] was used to perform the basic tree manipulation, the {phytools} package [76] was used to calculate Pagel's λ and Blomberg's K, the {caper} package [77] was used to perform phylogenetic generalized least squares (PGLS) regressions, and the {evomap} package [78] was used to calculate 95% confidence intervals and predictions intervals for the PGLS regressions. The resulting regressions of hyperpallium volume on Wulst surface area and of optic-tectum volume on optic-lobe surface area were used in the present analyses.

For our comparative methods, we used a tree that included all of the extant species in our sample as well as all of the extinct species for which brain-structure volume would be predicted. To generate this tree, we initially made a tree of extant birds by combining the backbone from Prum et al. [60] with the finer-level topology of Jetz et al. [61] as per the methods described in [79] using the R packages {ape} [75] and {phytools} [76]. This tree was reduced to only include the extant taxa in our sample with the "treedata" function in the {ape} package, then imported into Mesquite [80] where the extinct birds were added to the tree in the appropriate locations based on phylogenetic analyses from the literature [21,81–85]. For each fossil, the age of the resulting node was automatically assigned by Mesquite by dividing in half the length of the branch to which the fossil was added. This approach is repeatable and evenly distributes the influence of the fossil between the nodes above and below its attachment site (for published examples of this approach, see [86,87]). As a given fossil was being used to represent its species, the terminal age for the fossil was calculated as the midpoint of the range of its species. The branch length to the fossil was then calculated by subtracting its terminal age from the node age it was assigned by Mesquite and the branch length was adjusted accordingly in Mesquite. For the analyses of each trait, this combined tree of extant and extinct birds was reduced with the {ape} package to only include the taxa for which the trait of interest was available.

Previous work established that there is a strong, significant relationship between the surface areas of the Wulsts and optic lobes and their respective underlying brain structures in extant birds [51]. The surface areas of the Wulsts and optic lobes of the extinct birds in the sample were combined with a dataset of these endocast structures and their underlying brain structures in extant birds for analysis with the {BayesModelS} package [88]. R code to perform this analysis was adapted from [89]. The package {BayesModelS} conducts phylogenetic predictions using Bayesian inference to calculate the value of a trait that cannot be directly measured in the taxon of interest (e.g., the brain-structure volume

of an extinct bird). The calculations are based on a PGLS regression of the trait that cannot be measured in the taxon of interest, such as the brain-structure volume, on a trait that can be measured, such as the endocast-structure surface area. The analysis can only calculate one missing trait at a time; that is, one run was required for each extinct bird in the sample. In each run, the Wulst surface area of the fossil specimen of interest was added to the dataset of Wulst surface areas and hyperpallium volumes of the extant species. The tree generated in Mesquite representing the complete extant and extinct sample was imported and limited to include the 26 species of extant birds for which we measured the Wulst surface areas and to exclude all fossils but the one being tested with the {ape} package. The {BayesModelS} package drew on this dataset, this tree, and the Pagel's λ and Blomberg's K values for hyperpallium volumes stored from the extant-only analysis to calculate a predicted hyperpallium volume for the fossil in question. The analysis was set up with a burn-in of 20,000 iterations with sampling occurring every 50 iterations for 200,000 iterations. One run was performed for each of the five fossils for which the Wulsts were sufficiently preserved to be measured. The same approach was taken with a sample of 34 species of extant birds to calculate the optic-tectum volumes from optic-lobe surface areas for the seven extinct birds in the sample. The mean brain-structure volume for each run was recorded for subsequent analysis (Table S1).

To determine if any extinct taxa had brain-structure volumes that were significantly different from the extant birds studied, their predicted mean brain-structure volume and brain-remainder volume (endocast volume, as a proxy for brain volume, minus brain-structure volume) were compiled. These values were added to a dataset of actual brain-structure volume and brain-remainder volume (brain volume minus brain-structure volume) of all extant taxa for which these values were available in the literature [51,52,63–71] (Table S1). These values were also added to a second dataset that included endocast-structure surface area, endocast-remainder surface area, and foramen magnum metrics, although this dataset was limited to the extinct and extant birds for which endocasts were available.

The brain-structure volumes were regressed on brain-remainder volumes and phylogenetic ANCOVA (phyANCOVA) were performed with the R package {evomap} [78] using R code adapted from [90]. A phyANCOVA can be used to determine if a single species deviates from allometric predictions for a group by testing for differences in intercept using an estimate of the common slope calculated for the complete sample [56]. In other words, the slope of the line resulting from the regression of the traits of interest in the sample is calculated, then a line with the same slope is plotted through the data point representing the fossil of interest and the resulting intercept is calculated. This intercept is compared to the intercept of the line through the complete sample, representing a direct test of differences, unlike other approaches [56]. A phyANCOVA was performed using the predicted hyperpallium volumes for the five fossil specimens with quantifiable Wulsts and the predicted optic-tectum volumes for each fossil specimen, each time testing if one of the fossils was significantly different from the rest of the sample. The same sets of phyANCOVA were run on regressions of the endocast-structure surface areas on endocast-remainder surface areas, endocast-structure surface areas on foramen-magnum area and foramen-magnum width, and predicted brain-structure volumes on foramen-magnum area and foramen-magnum width. As this approach involves multiple testing, the *p*-values from all phyANCOVA were adjusted with the Benjamini–Hochberg procedure [91] in the {stats} package [73] to control for the rate of false discovery

Permission to share CT scan data and endocasts was obtained from curatorial staff and data contributors for as many specimens as possible. The image and shape files for these specimens are available on MorphoSource (Table S2). The tre file used for our phylogenetic comparative methods is available in the Supplementary Materials (S4), and R code used to run analyses will be available at https://github.com/cmearly/Beyond-Endocasts-R-Code.

## 3. Results

Wulst surface areas were measured and hyperpallium volumes were predicted for the five of the available fossil specimens that did not exhibit excessive deformation on the dorsal surface of their

endocasts (Table 2). The phyANCOVA tests performed on PGLS regressions of different combinations of hyperpallium size metrics and size correction factors indicated that none of the extinct birds in the sample had significantly different intercepts from the rest of the sample when slope was held constant (Table 3; Figure 4). In other words, neither the Wulst surface areas nor hyperpallium volumes relative to any of the metrics of brain size of any of the five extinct birds analyzed were significantly different from these values in the combined sample of extant and extinct birds.

**Table 2.** Measured Wulst surface areas and phylogenetically predicted hyperpallium volumes for all extinct specimens in the sample with quantifiable Wulsts.

| Specimen | Wulst Surface Area (mm$^2$) | Hyperpallium Volume (mm$^3$) |
|---|---|---|
| *Dinornis* | 1565.6 | 4044.3 |
| Miocene galliform | 127.9 | 227.1 |
| *Paraptenodytes* | 876.3 | 3116.7 |
| *Psilopterus* | 856.3 | 2316.1 |
| *Llallawavis* | 1640.6 | 5110.5 |

**Table 3.** Resulting *p*-values of the phylogenetic ANCOVA comparing the intercept of each extinct member of the sample to the intercept of the rest of the sample, given a common slope. These *p*-values have been adjusted with the Benjamini and Hochberg [89] procedure to control for false discovery rate; unadjusted *p*-values are available in Table S3. Abbreviations: BR, brain-remainder volume (brain volume minus optic-tectum volume); ER, endocast-remainder surface area (endocast surface area minus optic-lobe surface area); FMa, foramen-magnum area; FMw, foramen-magnum width; H, hyperpallium volume; W, Wulst surface area.

| Specimen | H v. BR | W v. ER | W v. FMa | W v. FMw | H v. FMa | H v. FMw |
|---|---|---|---|---|---|---|
| *Dinornis* | 0.9 | 0.75 | 0.98 | 0.69 | 0.95 | 0.91 |
| Miocene galliform | 0.9 | 0.92 | 0.98 | 0.5 | 0.95 | 0.8 |
| *Paraptenodytes* | 0.9 | 0.92 | 0.98 | 0.74 | 0.95 | 1 |
| *Psilopterus* | 0.9 | 0.92 | 0.98 | 0.69 | 0.95 | 0.8 |
| *Llallawavis* | 0.9 | 0.92 | 0.98 | 0.69 | 0.95 | 0.8 |

The optic-lobe surface areas were measured, and optic-tectum volumes were predicted for seven fossil specimens (Table 4). The measured optic-lobe surface areas and predicted optic-tectum volumes were regressed on various brain-size correction factors, and phyANCOVA were performed on these regressions (Figure 5). The two species that were consistently identified as outliers, the North Island brown kiwi (*Apteryx australis mantelli*) and the South Island giant moa (*Dinornis robustus*) are labeled on the plots (Figure 5). When the complete sample of extant and extinct birds was analyzed together, only the optic-tectum volume relative to brain remainder-volume of *Dinornis* was significantly smaller. In other words, in that regression, the intercept of a hypothetical line passing through the data point representing *Dinornis* when the line was assigned a common slope based on the rest of the sample was significantly different from the intercept of the sample (Table 5; Figure 5). However, when the outlier *Apteryx* was excluded, *Dinornis* was significantly different in all combinations of optic-tectum and optic-lobe size metrics and size correction factors (Table 6). No other extinct birds in the sample had significantly different relative optic-tectum or optic-lobe values regardless of whether *Dinornis* was included (Tables 6 and 7; Figure 5).

**Table 4.** Measured optic-lobe surface areas and phylogenetically predicted optic-tectum volumes for all extinct species in the sample.

| Specimen | Optic Lobe Surface Area (mm$^2$) | Optic Tectum Volume (mm$^3$) |
|---|---|---|
| *Archaeopteryx* | 115.2 | 106.4 |
| *Lithornis* | 172.6 | 184.9 |
| *Dinornis* | 194.3 | 237.64 |
| Miocene galliform | 172.9 | 205.5 |
| *Paraptenodytes* | 454.0 | 898.4 |
| *Psilopterus* | 454. 5 | 774.3 |
| *Llallawavis* | 664.3 | 1368.1 |

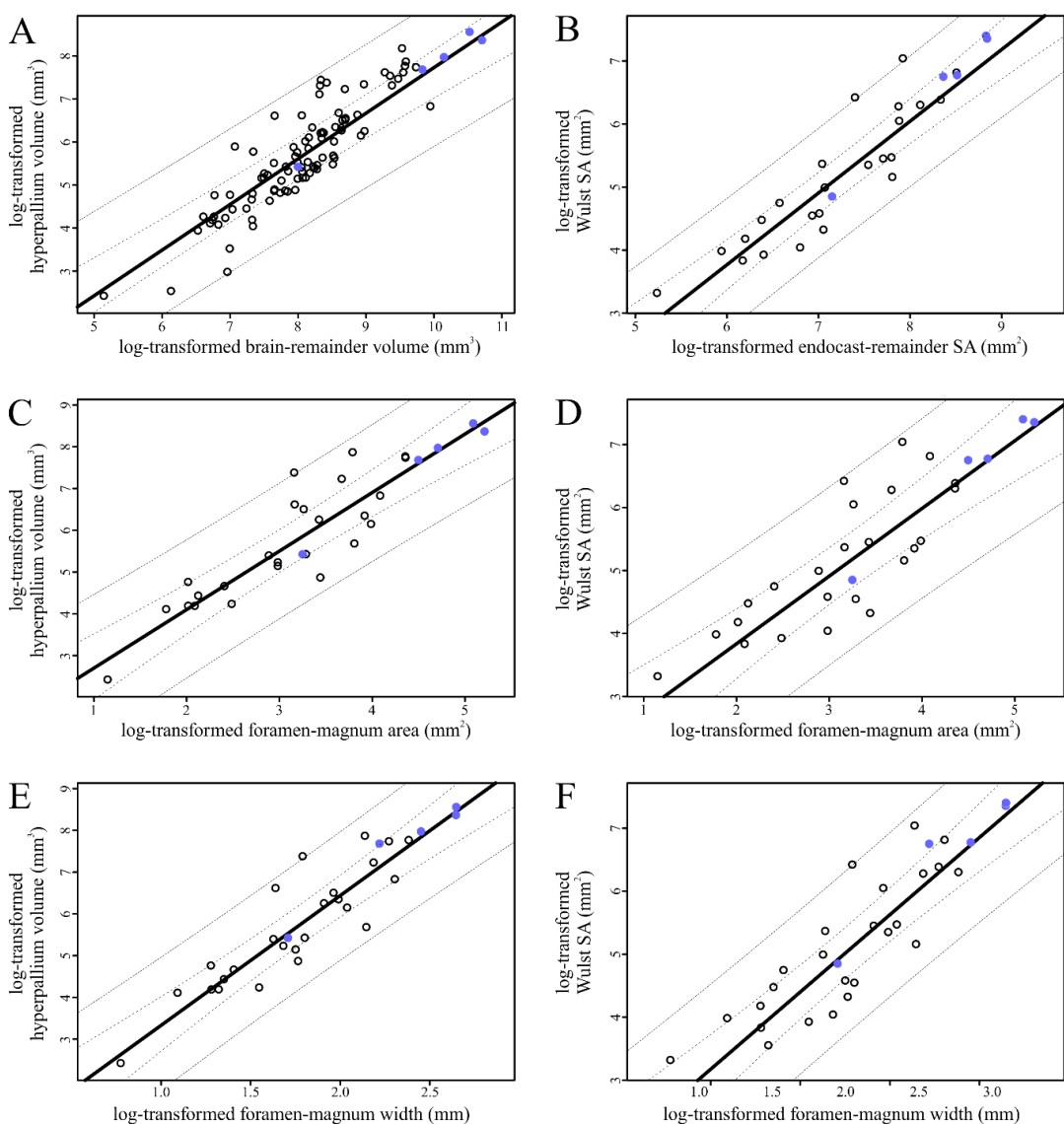

**Figure 4.** Scatterplots of hyperpallium metrics. All values are log-transformed, and on each plot, the solid line represents a phylogenetic generalized least squares (PGLS) regression from the extant sample, the dotted lines represent confidence intervals, and the dashed lines represent prediction intervals. The open points represent extant taxa. The values for the extinct sample are plotted as blue points. (**A**) Hyperpallium volume regressed on brain-remainder volume. (**B**) Wulst surface area regressed on endocast-remainder surface area. (**C**) Hyperpallium volume regressed on foramen-magnum area. (**D**) Wulst surface area regressed on foramen-magnum area. (**E**) Hyperpallium volume regressed on foramen-magnum width. (**F**) Wulst surface area regressed on foramen-magnum width.

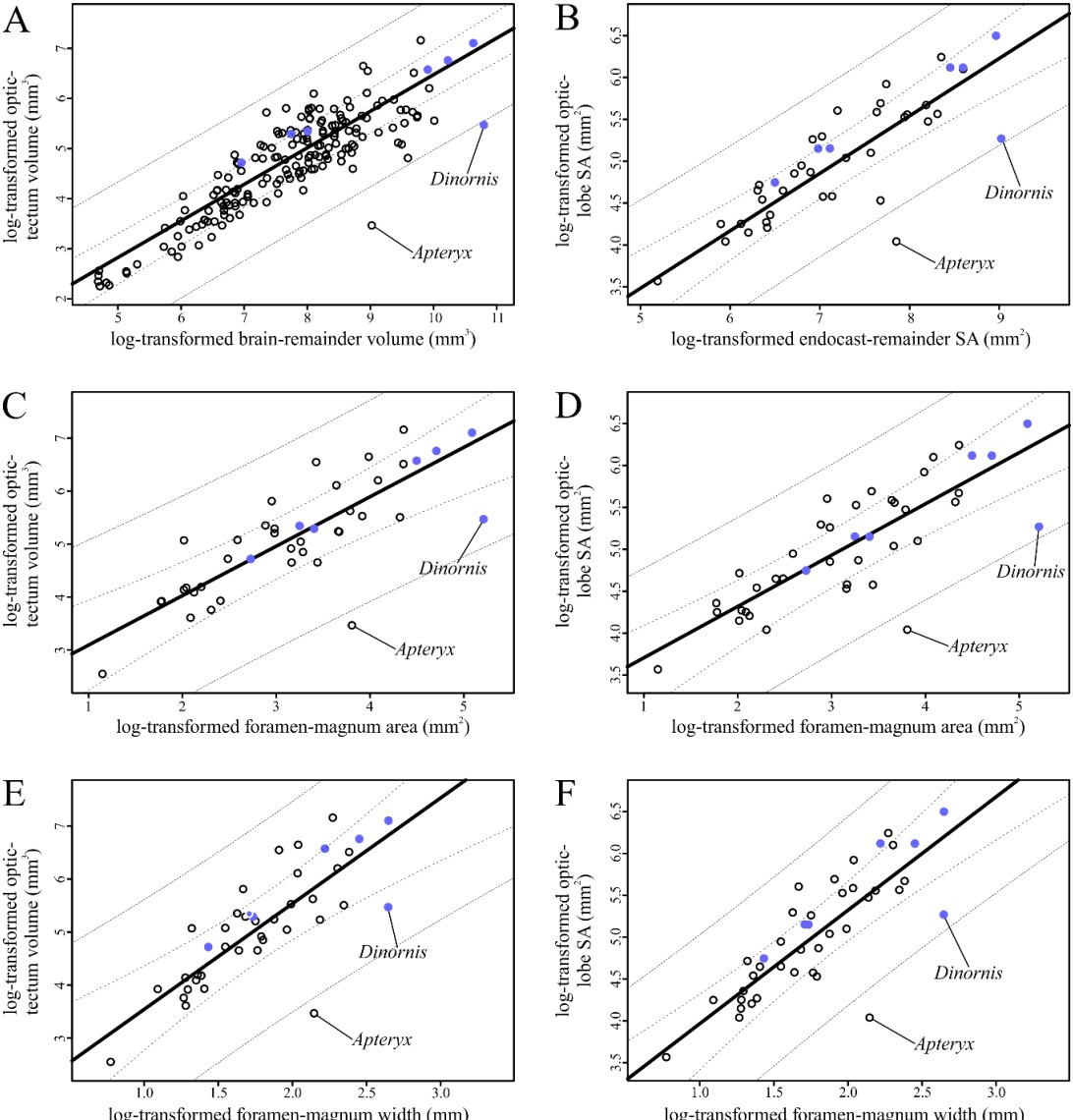

**Figure 5.** Scatterplots of optic-tectum metrics. All values are log-transformed, and on each plot, the solid line represents a PGLS regression from the extant sample, the dotted lines represent confidence intervals, and the dashed lines represent prediction intervals. The open points represent extant taxa. The values for the extinct sample are plotted as blue points and the points representing *Dinornis* and *Apteryx* are indicated on each plot. (**A**) Optic-tectum volume regressed on brain-remainder volume. (**B**) Optic-lobe surface area regressed on endocast-remainder surface area. (**C**) Optic-tectum volume regressed on foramen-magnum area. (**D**) Optic-lobe surface area regressed on foramen-magnum area. (**E**) Optic-tectum volume regressed on foramen-magnum width. (**F**) Optic-lobe surface area regressed on foramen-magnum width.

**Table 5.** Resulting *p*-values of the phylogenetic ANCOVA comparing the intercept of each extinct member of the sample to the intercept of the rest of the sample, given a common slope. These *p*-values have been adjusted with the Benjamini and Hochberg [89] procedure to control for false discovery rate; unadjusted *p*-values are available in Table S3. Significant differences are highlighted in aqua. Abbreviations: BR, brain-remainder volume (brain volume minus optic-tectum volume); ER, endocast-remainder surface area (endocast surface area minus optic-lobe surface area); FMa, foramen-magnum area; FMw, foramen-magnum width; OL, optic-lobe surface area; OT, optic-tectum volume.

| Specimen | OT vs. BR | OL vs. ER | OL vs. FMa | OL vs. FMw | OT vs. FMa | OT vs. FMw |
|---|---|---|---|---|---|---|
| *Archaeopteryx* | 0.95 | 0.64 | 0.97 | 0.51 | 0.98 | 0.71 |
| *Lithornis* | 0.30 | 0.64 | 0.99 | 0.73 | 0.98 | 0.71 |
| *Dinornis* | $3.5 \times 10^{-3}$ | 0.11 | 0.084 | 0.15 | 0.07 | 0.19 |
| Miocene galliform | 0.37 | 0.90 | 0.97 | 0.73 | 0.98 | 0.71 |
| *Paraptenodytes* | 0.95 | 0.67 | 0.97 | 0.73 | 0.98 | 0.71 |
| *Psilopterus* | 0.95 | 0.64 | 0.97 | 0.51 | 0.98 | 0.71 |
| *Llallawavis* | 0.95 | 0.64 | 0.97 | 0.73 | 0.98 | 0.71 |

**Table 6.** Resulting *p*-values of the phylogenetic ANCOVA comparing the intercept of each extinct member of the sample to the intercept of the rest of the sample, given a common slope, with the outlier *Apteryx* excluded. These *p*-values have been adjusted with the Benjamini and Hochberg [89] procedure to control for false discovery rate; unadjusted *p*-values are available in Table S3. Significant differences are highlighted in aqua. Abbreviations: BR, brain-remainder volume (brain volume minus optic-tectum volume); ER, endocast-remainder surface area (endocast surface area minus optic-lobe surface area); FMa, foramen-magnum area; FMw, foramen-magnum width; OL, optic-lobe surface area; OT, optic-tectum volume.

| Specimen | OT vs. BR | OL vs. ER | OL vs. FMa | OL vs. FMw | OT vs. FMa | OT vs. FMw |
|---|---|---|---|---|---|---|
| *Archaeopteryx* | 0.92 | 0.77 | 0.91 | 0.56 | 0.9 | 0.72 |
| *Lithornis* | 0.56 | 0.56 | 0.92 | 0.73 | 0.9 | 0.73 |
| *Dinornis* | $3.4 \times 10^{-4}$ | $3.4 \times 10^{-4}$ | $9.1 \times 10^{-3}$ | $4.9 \times 10^{-3}$ | $4.9 \times 10^{-3}$ | $8.4 \times 10^{-3}$ |
| Miocene galliform | 0.56 | 0.56 | 0.92 | 0.73 | 0.9 | 0.73 |
| *Paraptenodytes* | 0.94 | 0.77 | 0.92 | 0.73 | 0.9 | 0.73 |
| *Psilopterus* | 0.92 | 0.77 | 0.92 | 0.56 | 0.9 | 0.72 |
| *Llallawavis* | 0.92 | 0.77 | 0.92 | 0.73 | 0.9 | 0.73 |

**Table 7.** Resulting *p*-values of the phylogenetic ANCOVA comparing the intercept of each extinct member of the sample to the intercept of the rest of the sample, given a common slope, with the outliers *Apteryx* and *Dinornis* excluded. These *p*-values have been adjusted with the Benjamini and Hochberg [89] procedure to control for false discovery rate; unadjusted *p*-values are available in Table S3. Abbreviations: BR, brain-remainder volume (brain volume minus optic-tectum volume); ER, endocast-remainder surface area (endocast surface area minus optic-lobe surface area); FMa, foramen-magnum area; FMw, foramen-magnum width; OL, optic-lobe surface area; OT, optic-tectum volume.

| Specimen | OT vs. BR | OL vs. ER | OL vs. FMa | OL vs. FMw | OT vs. FMa | OT vs. FMw |
|---|---|---|---|---|---|---|
| *Archaeopteryx* | 0.97 | 0.71 | 1 | 0.89 | 1 | 0.94 |
| *Lithornis* | 0.97 | 0.71 | 1 | 0.91 | 1 | 0.94 |
| Miocene galliform | 0.97 | 0.71 | 1 | 0.91 | 1 | 0.94 |
| *Paraptenodytes* | 0.97 | 0.71 | 1 | 0.89 | 1 | 0.94 |
| *Psilopterus* | 0.97 | 0.71 | 1 | 0.89 | 1 | 0.94 |
| *Llallawavis* | 0.97 | 0.71 | 1 | 0.89 | 1 | 0.94 |

## 4. Discussion

We used phylogenetic prediction methods to calculate brain-structure volumes from endocast-structure surface areas of a sample of extinct birds with relatively undeformed endocasts.

We performed phyANCOVA on regressions of these predicted brain-structure volumes and the measured endocast structure surface areas on various metrics of brain size, including foramen magnum size. In general, our results suggest that most of the extinct birds studied did not differ significantly from their extant relatives in the size of their hyperpallia (or overlying Wulsts) or their optic tecta (or overlying optic lobes). The exception to this trend is the specimen of *Dinornis robustus*, which had a significantly smaller optic tectum relative to its brain size. The relative reduction of the optic tectum of *Dinornis*, a specimen with a completely undeformed braincase, likely has functional implications for its visual abilities, discussed further below.

It is important to discuss the potential impacts of deformation on the estimates of brain size in extinct birds, but the present sample comprises endocasts with as little deformation as possible. The degree of deformation in the sample exists on a spectrum, ranging from none in *Dinornis* to deformation so severe in one hemisphere of *Archaeopteryx* that the less-deformed hemisphere had to be mirrored (Figure 2). Breakage is evident in all specimens except for *Dinornis*, but all of the breaks present in the fossil sample resulted in minimal offset between the bone fragments. Because of the relatively good preservation of the fossils in this sample, minimal efforts were made at retrodeformation. Holes and breaks were patched in a manner that resulted in smooth transitions that reflected general avian brain shape for the given region, and hemispheres were mirrored across the sagittal midline when appropriate. Given the recent efforts to develop rigorous methods for retrodeformation of vertebrate fossils [92–94], avian skulls with more severe damage may be incorporated into these analyses in the future.

We attempted to account for the minimal deformation present in the studied endocasts by including foramen-magnum area and width as a size metric, which is a novel approach in avian endocast studies. Comparative histological studies have used brain-remainder volume to examine how discrete brain structures scale compared to the rest of the system in extant birds. Unfortunately, the skulls of extinct birds have often been subjected to at least partial deformation in the form of crushing, resulting in endocasts that may be volumetrically smaller than the endocast (and brain) would have been during life. Because the foramen magnum width often showed minimal or no deformation in our sample of fossils, it was used as an alternative scaling factor to supplement endocast-remainder volume. Foramen-magnum area was also included as it should provide a more inclusive estimate of the cross-sectional area of the brainstem and spinal cord within the foramen magnum. Examining how brain-structure volumes scale to foramen-magnum size has been done in other groups of animals [72], but is not commonly practiced with avian datasets. To fully establish it as an acceptable scaling factor for relative brain-structure size studies, a large dataset of endocranial volumes and foramen-magnum size metrics should be analyzed for scaling variation among taxa. In the absence of such a study, regressions of brain-structure volume on foramen-magnum area or width should be interpreted with caution. Still, such regressions are useful to supplement regressions of brain-structure volume on brain-remainder volume when brain-remainder volume is measured from deformed endocasts.

The application of phyANCOVA to these datasets represents a new approach to analyzing fossil endocasts. Other well-established methods of testing samples, such as calculating confidence and prediction intervals of a phylogenetic regression, do not constitute a direct test and thus do not allow users to evaluate if the difference is significant [56]. Publications on relative brain-structure size in extant species based on histology that have identified hyper- or hypotrophy of structures in certain groups and tied these trends to functional implications have been based on group means e.g., [52]. However, unlike in extant samples where each sampled clade comprises multiple species, group means cannot usually be calculated for fossil specimens, where the presence of even one 3D-preserved skull of a given species of bird is an extremely rare occurrence. The application of Smaers and Rohlf's phyANCOVA approach [56], aimed specifically at testing if one species is significantly different from the rest of the group, provides a new way to incorporate fossils into quantitative comparative neuroanatomical studies.

Based on the results of our phyANCOVA, none of the extinct birds sampled were significantly different in terms of Wulst or hyperpallium size compared to the other birds sampled (Figure 3; Table 3). The fact that none of the Wulsts or hyperpallia of the extinct birds were larger than those of their extant relatives was unsurprising as only owls and some caprimulgiforms have been found to have relatively larger hyperpallia and no fossil representatives of these groups were studied [66]. The lack of relatively smaller Wulsts and hyperpallia in the extinct sample may reflect a preservation bias in the sample toward more recently extinct birds. The Wulst, as the externally-visible expression of the hyperpallium on the dorsal surface of the brain, is thought to have evolved in the early Cenozoic or even Late Cretaceous [20], and the oldest fossil in our sample for which Wulst surface area could be measured was the 22-million-year-old *Paraptenodytes* [21]. Thus, it is likely that our extinct sample does not capture the period in avian evolution during which the hyperpallium was expanding to the size seen in most birds today. The hyperpallium is also hypothesized to have been key to avian diversification and success after the K-Pg extinction [14,19], so there may be selective pressure to maintain a minimum size of this avian brain structure.

Although the endocast of one Mesozoic bird, *Archaeopteryx*, was available for study, we elected not to include it in our analyses of Wulst and hyperpallium size and instead just include it in our analyses of optic lobe and optic tectum size. A large putative Wulst was identified on the endocast of the London specimen of *Archaeopteryx* (NHMUK PAL PV OR 37001, the same specimen from which the endocast in the present study was generated) which moved the origin of the Wulst to a much earlier point in maniraptoran evolution than previously hypothesized [10]. Walsh et al. [48] argued that the lack of a Wulst in the Cretaceous bird *Cerebravis* and the weak development of the Wulsts present on the endocasts of some early Cenozoic birds [19,20] do not support the interpretation of a Wulst in *Archaeopteryx*. Recently, Beyrand et al. [14] subjected the London specimen and another specimen of *Archaeopteryx* (the Munich specimen) to synchrotron propagation phase-contrast X-ray microtomography, revealing that the putative Wulst feature in the London specimen is in fact a fracture and that the corresponding but unbroken surface of the Munich specimen shows no evidence of the Wulst.

Most of the extinct birds sampled fell within the range of variation in optic-lobe and optic-tectum size represented in the extant sample (Figure 5; Tables 5–7). The exception to this trend is *Dinornis*, which has a smaller optic tectum relative to the brain size (Figure 5; Table 5) when compared to the rest of the sampled birds. When the outlier *Apteryx* was removed from the sample, the difference between *Dinornis* and the rest of the sample was significant regardless of which size metric for the structure of interest (optic-tectum volume or optic-lobe surface area) was regressed on which brain-size metric (brain-remainder volume, endocast-remainder surface area, or foramen-magnum area or width; Table 6). The significant reduction of the optic tectum relative to foramen-magnum size indicates that this is a true reduction rather than a possible artifact of expanded cerebral hemispheres, as has been suggested in other birds [52]. These results support earlier hypotheses of reduction of the optic lobe based on the qualitative evaluations of the endocast [49–51]. The extreme reduction of this structure is unique to *Dinornis* in our sample of extinct birds (Figure 5; Tables 5–7).

Results of our previous study indicate that variation in the size of the optic lobe explains the majority of the variation in the size of the optic tectum [51]. This means that the reduced optic lobe of *Dinornis* likely represents a reduction in the optic tectum. Following the Principle of Proper Mass [4], which states that the relative size of a brain structure correlates with the relative importance to the animal of the information processed by that structure, this reduction could indicate reduced emphasis on visual information. The only specimen in our sample that shows a more extreme reduction in these structures is the North Island brown kiwi (*Apteryx australis mantelli*), which is a palaeognath like *Dinornis*, although Apterygiformes are not sister to Dinornithiformes. Kiwi are flightless, nocturnal birds from New Zealand that have relatively smaller optic tecta and relatively larger brain structures that are devoted to processing olfactory information and tactile information from the bill tip [95]. The reduction of the visual system in this genus is so pronounced that some

individuals in a wild population of kiwi are blind [96]. Other aspects of the cranial anatomy of *Dinornis* such as the overall endocast morphology (Figures 6 and 7) [18], bill shape, and orbit size [97] do not support *Dinornis* having the same extreme sensory ecology as kiwi, but the relative size of its optic lobe is still remarkably small compared to other birds.

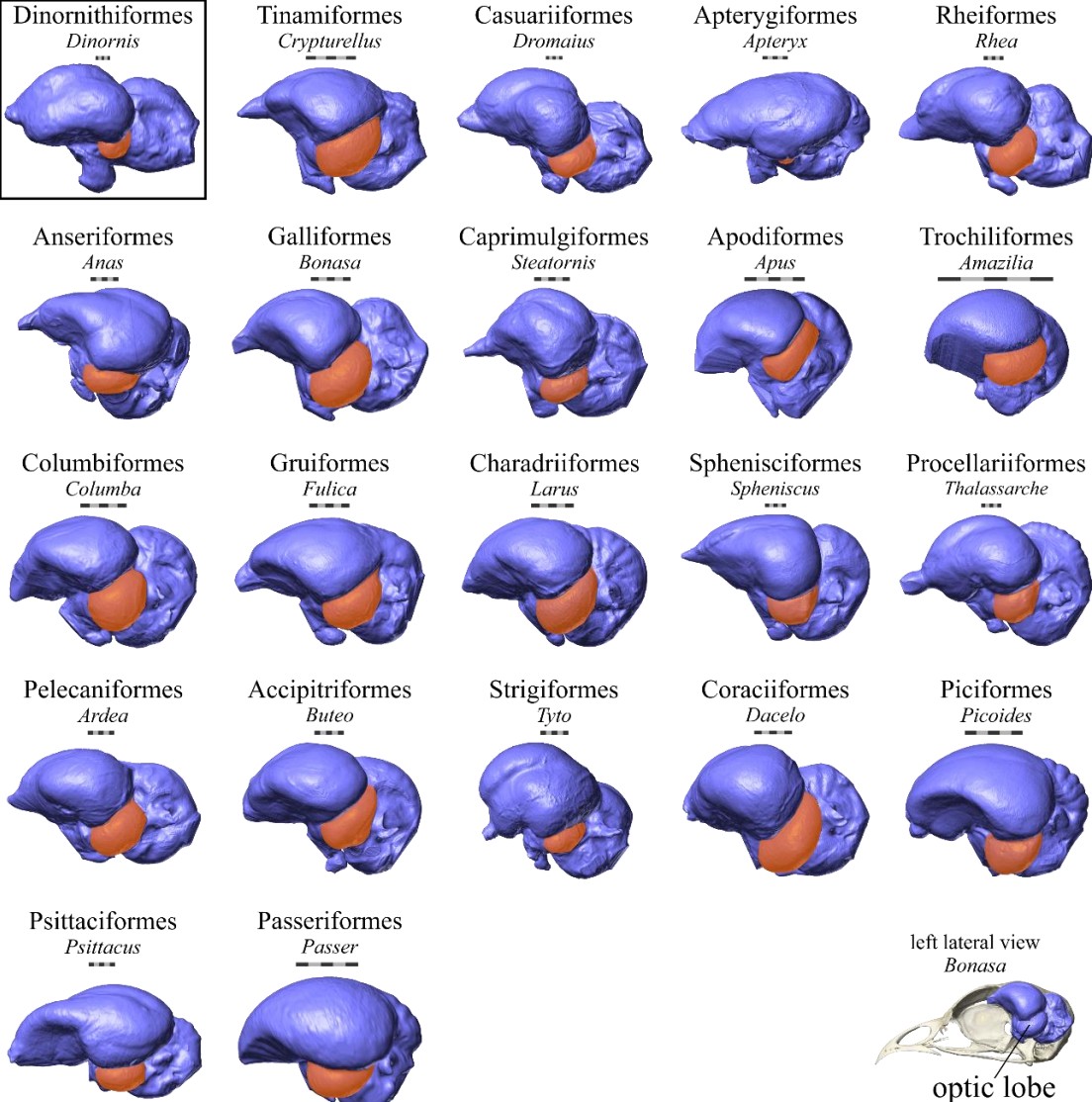

**Figure 6.** The endocast of *Dinornis robustus* and a representative sample of endocasts of the extant orders of birds analyzed in this study in left lateral view (see skull and endocast of *Bonasa umbellus* at bottom right for orientation). Optic lobes are highlighted in orange to aid in comparison of relative size of this structure across taxa. Scale bars = 5 mm.

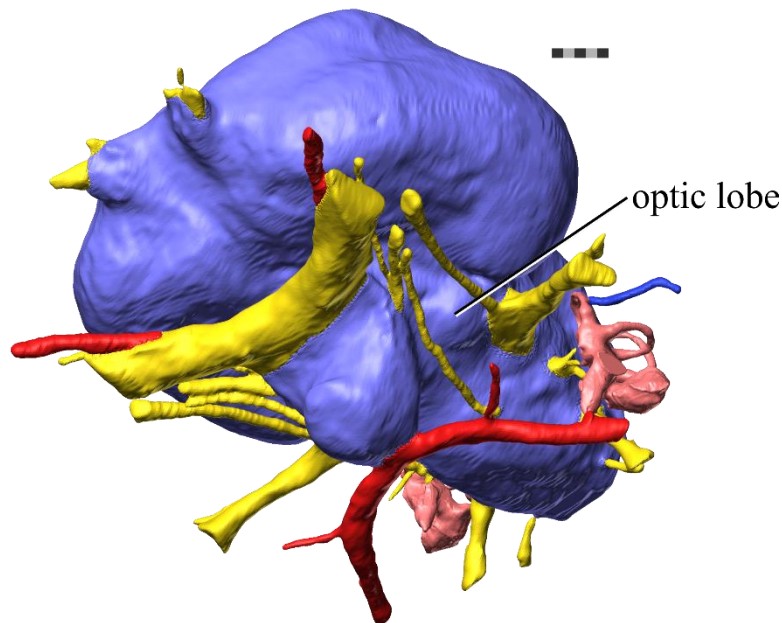

**Figure 7.** The endocast of *Dinornis robustus* in a left ventral oblique view to highlight the size and position of the optic lobe relative to other brain structures. The brain endocast is modeled in light blue, cranial nerves are modeled in yellow, arteries are modeled in red, veins are modeled in dark blue, and semicircular canals are modeled in pink. Scale bar = 5 mm.

It is important to note that, although it is most likely that the reduction in the optic lobe is correlated with a reduction in the optic tectum of *Dinornis*, there are other structures housed in the optic lobe that could contribute to its size. One of these is the nucleus lentiformis mesencephali (LM), which processes the movement of images across the retina as part of the optokinetic response [98]. The components of the optokinetic response use the information on movement through space processed by LM to produce compensatory movements of the eyes and the head to stabilize the image on the retina [98,99]. Whether the reduction of the optic lobe represents a reduction of the optic tectum, LM, both structures, or other structures, a relatively smaller optic lobe suggests that *Dinornis* had reduced visual abilities compared to extant birds. Further implications of the endocranial anatomy and other anatomical traits related to the visual system of this specimen of *Dinornis* such as the orbit and the optic nerve will be discussed in detail in a forthcoming publication dedicated to this specimen.

The fact that the sizes of the hyperpallia, Wulsts, optic tecta, and optic lobes of this sample of extinct birds generally fall within the variation documented in extant birds lends further support to these structures being considered hallmarks of avian brain evolution [7]. It may also indicate a tendency of these structures to change in size in concert with overall brain size, although this was not tested directly. *Dinornis* breaks these trends by having a relatively smaller optic tectum and optic lobe than expected given the optic tectum volumes and optic lobe surface areas of our extant and extinct sample. This secondary reduction in size is not unheard of as it has also been documented quantitatively in some extant birds [52,67,95] and qualitatively in some extinct birds [9,22,26], but it does have potential implications for the sensory abilities of *Dinornis*. In addition to the potential functional and evolutionary implications of our findings for the extinct birds in our sample, our findings serve as case studies for a novel approach to studying the evolution of neuroanatomy in birds. We provide a quantitative framework for predicting and testing for differences in the sizes of two vision-related brain structures in the fossil record, further extending our understanding of avian brain evolution into deep time.

## 5. Conclusions

This study presents a quantitative, replicable workflow for predicting the volumes of the optic tecta and hyperpallia of extinct birds and identifying extinct birds in which these brain structures are significantly different in size from those of a broad sample of birds. Phylogenetic prediction methods allow incorporation of extinct taxa into a large comparative dataset of brain-structure size. The resulting focus on brain-structure size in extinct birds strengthens the functional inferences that can be made from fossils, as the sizes of the structures that have been correlated with sensory abilities (e.g., hyperpallium, optic tectum) can be directly studied in addition to their endocast proxies (e.g., Wulst, optic lobe). The phyANCOVA approach we employed provides a direct test of differences in these aspects of neuroanatomy in extinct birds and allows a clearer view of avian brain evolution. These methods could be applied to other published endocasts of extinct birds hypothesized to have relatively smaller Wulsts e.g., [19,20,23–25] or relatively smaller optic lobes e.g., [9,22,26] to test if the qualitative assessments of relative brain-structure size based on endocast morphology were borne out by these new quantitative methods.

**Supplementary Materials:** The following are available online at http://www.mdpi.com/1424-2818/12/1/34/s1, Table S1: brain and endocast measurements; Table S2: CT scanned specimens; Table S3: unadjusted *p*-values; S4: complete TRE file.

**Author Contributions:** Conceptualization, C.M.E. and L.M.W.; methodology, C.M.E., R.C.R. and L.M.W.; software, C.M.E.; validation, C.M.E.; formal analysis, C.M.E.; investigation, C.M.E.; resources, L.M.W.; data curation, C.M.E., R.C.R. and L.M.W.; writing—original draft preparation, C.M.E.; writing—review and editing, C.M.E., R.C.R. and L.M.W.; visualization, C.M.E. and R.C.R.; supervision, C.M.E. and L.M.W.; project administration, C.M.E.; funding acquisition, C.M.E., R.C.R. and L.M.W. All authors have read and agreed to the published version of the manuscript.

**Funding:** This research was supported by: The National Science Foundation (NSF) Graduate Research Fellowship Program under Grants No. DGE-1060934 and DGE-1645419 to C.M.E.; NSF Grants No. IOB-0517257, IOS-1050154, and IOS-1456503 to L.M.W. and R.C.R.; grants from the Ohio University (OU) Graduate Student Senate and Office of the Vice President for Research and Creative Activity to C.M.E.; and a Chapman Ornithology Grant from the Frank M. Chapman Memorial Fund of the American Museum of Natural History to C.M.E.

**Acknowledgments:** Discussions at "A Deeper Look Into the Avian Brain: Using Modern Imaging to Unlock Ancient Endocasts," a Catalysis Meeting convened at the National Evolutionary Synthesis Center (funded by NSF EF-0905606) by A. Balanoff, D. Ksepka, and N. A. Smith, were key in early development of this project. For further insight and discussion, we thank J. Bourke, D. Cerio, F. Degrange, A. Iwaniuk, A. Morhardt, J. Nassif, and W. R. Porter. Special thanks to A. Iwaniuk for extensive discussion and for providing access to histological brain data that were collected for other studies [51] and analyzed and re-published in this study. Previous versions of this manuscript were reviewed by D. Blackburn, S. Kuchta, P. O'Connor, and A. Stigall, and feedback from anonymous reviewers greatly improved this manuscript. We thank R. Felice, H. O'Brien, and D. Ksepka for assistance with statistical analyses and phylogenetic comparative methods, and A. Balanoff, F. Degrange, T. Gaetano, S. Jordan, D. Ksepka, A. Milner, W. R. Porter, and M. Wakui for contributions of CT scan data and endocast segmentation. CT scanning performed at University of Washington's Friday Harbor Labs was supported in part by NSF DBI 1701665. We are grateful to collections managers and curatorial staff at the American Museum of Natural History, the Carnegie Museum of Natural History, the Field Museum of Natural History, and the Smithsonian National Museum of Natural History for loans of specimens in their care.

**Conflicts of Interest:** The authors declare no conflict of interest. The funders had no role in the design of the study; in the collection, analyses, or interpretation of data; in the writing of the manuscript, or in the decision to publish the results.

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
