# Peer review of "Beyond Endocasts: Using Predicted Brain-Structure Volumes of Extinct Birds to Assess Neuroanatomical and Behavioral Inferences"

_diversity, doi:10.3390/d12010034_

Round 1
Reviewer 1 Report
This manuscript describes an analytical framework to evaluate whether imputed brain structure volumes in extinct birds deviate from allometric expectations relative to extant birds. The goal of the manuscript is described as providing a proof-of-concept for future studies.
The manuscript is well written and provides appropriate detail in justifying the goals of the study, as well as the materials and methods used.
One element that should be added to the statistical analysis is to control for false discovery rate in multiple testing. The authors perform multiple tests by testing whether each of five fossils significantly deviates from allometric predictions. This is an appropriate procedure given the research question. But because this constitutes multiple testing, a procedure to control for false discovery rate should be applied. I strongly suspect this will not alter the results given that the only instance of statistical ‘discovery’ is highly significant. Nonetheless, such a control is necessary to demonstrate the highest standard of rigor. I recommend the use of the Benjamin & Hochberg procedure (using the function ‘p.adjust’, and method ‘BH’ in the ‘stats’ R package).
Furthermore, I wonder which extant species is the outlier in the plots of Figure 4. Is there something particular about the profile of this bird that explains its lower than expected tectum volume? If so, this might inform what is happening with Dinornis. If not, is this measurement error? And if this is indeed measurement error, could such measurement error also be attributed to Dinornis?
Reviewer 2 Report
The paper entitled “Beyond endocasts: using predicted brain structure volumes of extinct birds to assess neuroanatomical and behavioral inferences” presents an interesting and novel method for analyzing volumes of brain structures from endocast surface areas of extinct avians. The paper in general, and especially the introduction, is well written, making it easy to follow for audiences of disparate fields. The methods are able to target information about internal brain anatomy that is usually lost in fossilized specimens, and therefore is of great importance to the blossoming avian paleoneuroanatomical field. The images and tables are clear and relevant to the text. I believe this study will be of interest for the audience of Diversity, and I recommend this paper for publication after very minor revisions, listed below by line number, with corresponding starts to relevant sentences in case the numbers shift.
Individual Comments:
- Lines 138-142(“These predicted brain structure volumes…”): I had a hard time following what was included in ‘brain structure volumes’ and ‘brain volume’, so I suggest adding a clause or parenthetical to indicate what those terms include and exclude, e.g. “brain structure volume (hyperpallium or optic tectum volume)”.
- Line 181(“Endocasts of seven…”): This sentence would make more sense at the beginning of the previous paragraph, which currently starts with “The endocasts of the terror birds….”. It seems out of place in its current location. The next sentence should then be edited to say “The skulls of each specimen were scanned…”.
- Paragraph starting at line 234(“The methods used to measure…”): Did you simultaneously add the faces to the brain structure and subtract them from the remainder? This is unclear from the description.
- Lines 286-288(“The branch length to the fossil…”): Why did you use this method for calculating branch length, especially with regard to using last occurrence and not use first occurrence? Please clarify in the text.
- Lines 408-410(“The relative reduction of the optic… likely has implications for its visual abilities”): Though I understand that you discuss this more further down in the section, I think it would be better to add a clause such as, “discussed further below” so as to not leave your readers on a cliff-hanger.
- Lines 518-520(“Another significant result…”): This sentence feels very awkward as the final sentence of the discussion, given that the sentence preceding it was about Dinornis. I also feel that this last sentence showcases the extremely significant aspect of this work – a method for analyzing internal components of endocasts, but because of the wording it seems like a secondary, less-significant conclusion. I think adding a sentence before this one to ease the transition into it, and also clarifying its importance will help.
- Figure 4:What’s the outlier to the bottom left of Dinornis? It would be helpful to indicate the species, and perhaps add a sentence or two describing potential reasons behind its position on these plots. It is clear from the text that you left in outliers to get the most accurate sample possible. It would be interesting to know more about this species.

Reviewer 3 Report
The manuscript is a solid, carefully executed piece of work that uses cutting-edge techniques for the phylogenetic analysis of fossil skull cavities (endocasts). The results are not earth-shattering but interesting insofar that they (a) confirm that the overall pattern of avian brain morphology was established a long tie ago and (b) reveal for the first time that the giant Moa (Dinornis) had an optic tectum that was significantly smaller than expected, based on the general avian pattern. As such, the results are worth publishing. I do, of course, have several suggestions for improvement.
Major Issues:
(1) I would have liked to see the endocast images earlier in paper, not as the last figure. More importantly, I really wanted to see a much better image of the Dinornis endocast, maybe even something like the youtube video of this same endocast I found on internet (from Witmer lab). It that rotating endocast, I could actually see the tectum and convince myself of its unusually small size. The small image in your figure 5 did not do it justice, especially since the “red paint” obscured what I was trying to see. I also recommend leaving the trigeminal nerve in the image or (better) movie. So, I recommend putting figure 5 first and then adding a figure/movie of Dinornis at the end. Something like that.
The current manuscript heavily relies on “previous work” that was, as far as I can tell, only published in a dissertation and an abstract. As a result, (2) I was often puzzled about the details of the “phylogenetic volume prediction” method. A better description of the method and, perhaps, an informative figure from that work would help. For example, I wondered: how good are the predictions? How much variation is there across species in the thickness of the wulst or tectum (and curvature of the structures’ surfaces), and how does the predictive model handle this variation?
(3) I missed an illustration of the phylogeny. Where do the extinct species fit, relative to the origin of all birds, relative to extant lineages, and relative to one another. Some of this information is in the text, but a figure (with divergence time estimates) would have been very helpful.
(4) In Table 5 I was startled to find that the optic lobe surface area for Dinornis is slightly larger than that for Lithornis, yet the predicted volume is much, much less. Assuming this is not a mistake, I suspect it has to do with your prediction being “phylogenetic” rather than a simple volume-from-surface formula, but I think you should explain this finding a little more, simply because it is startling (at least to me). Can we conclude that the optic tectum of Dinornis is much, much thinner than that of Lithornis?
(5) Lines 457-458: When you write that the hyperpallium evolved in the early Cenozoic, I suspect you mean that the Wulst as a delimitable structure on the brain surface evolved around that time. The hyperpallium itself is thought to be homologous to the “dorsal pallium” of other reptiles and mammals and is, therefore, “new in name only” with birds. Please clarify this.
(6) I agree that your data suggest Dinornis might have reduced the size of its visual circuitry, but what about eye/orbit size in this species? Is that also smaller than expected? If not, then something even more interesting must be going on. Please include at least a mention of eye size and whether it supports your hypothesis.
Minor Issues:
line 64: Please replace “behavior” with “functions”
lines 80-81: I don’t think you can really exclude the possibility of a change in structure size simply because the lineages diverged too recently. I think you’re making an unjustified/unsubstantiated assumption about the constancy of rates of evolutionary change in brain region size. I wouldn’t.
line 158: delete “criteria of”
line 192: “where these structures met”; met what?
lines 203-204: the issue of the trigeminal nerve is especially important for the Dinornis specimen, and I encourage you to include it in the endocast image I am requesting. On the video, I felt that I could see the tectum pretty well under the nerve.
line 208: I think “often-fragmented” can be two words.
lines 212-14: Why single out Ridgley, and not just say “we”? At first I didn’t know who “Ridgley” was/is.
lines 254-255: I agree that most avian endocasts reflect the brain surface quite well. However, I am not certain that the fit between brain and endocranial cavity would be equally tight in small birds vs large birds. Across vertebrates, the fit is usually looser in larger species. Is this worth addressing? Especially with regard to your largest specimens? I am not sure.
lines 272-3: If you have three Lithornis specimens but the foramen magnum is damaged in one, does it really make sense to average foramen magnum size across all three specimens? I would think discarding the broken one is better.
Paragraph starting at line 275: This whole paragraph was very confusing to me. How many trees are you working with? Please rewrite with more effort at clarity.
lines 315-16: It would have been great if you had determined “predicted brain volume” in the same way you determined “brain structure volume” rather than simply using endocast volume, but perhaps that isn’t necessary (assuming the endocast tracks the brain surface tightly). In any case, I’d put commas before and after “as a proxy for brain volume” to minimize ambiguity.
lines 348-349: Unless readers are familiar with the Smaers/Rohlf paper, they might be confused about how you can get an intercept for a single data point. It would help to explain that you are determining this intercept using a line through your single data point with the same (“common”) slope as the line that describes the entire data set (minus your data point?). You might even try to explain why this is better than simply looking at residuals and asking whether they fall outside of the prediction interval, etc. It’s not an easy methodology for most readers to grasp, I suspect.
Figures 3&4: why do we need panels E and F? Why look at foramen magnum width separately from FM area? I don’t think you told me…
Figure 4: which species is the other outlier in the graphs (i.e., the species that deviates just about as far from the other species as Dinornis does)? Is it a close relative of Dinornis?
lines 464+: I was puzzled that you here explain why you excluded Archaeopterix from your analysis, when it was included in your table 1. Did you include it just for the optic lobe analyses? It would help to clarify.
lines 499-500: I would write “importance to the animal”, though I don’t remember Jerison’s precise wording. “Important” is in the eye of the beholder.
Round 2
Reviewer 3 Report
I appreciate how carefully the authors have responded to my concerns. My only remaining concern is that the graph figures have lost their resolution (compared to the graphs submitted in version #1), as well as their confidence and prediction intervals. I'm glad to see that Apteryx is now labeled, but the original figure quality should be restored.